# Assessing the Respect of Children’s Rights in Pediatric Hospitals

**DOI:** 10.3390/medicina59050955

**Published:** 2023-05-16

**Authors:** Vasiliki Georgousopoulou, Antonis Voutetakis, Petros Galanis, Freideriki Eleni Kourti, Afroditi Zartaloudi, Ioannis Koutelekos, Evangelos Dousis, Dimitrios Kosmidis, Sotiria Koutsouki, Despoina Pappa, Michael Igoumenidis, Chrysoula Dafogianni

**Affiliations:** 11st Department of Pediatrics, Medical School, National and Kapodistrian University of Athens, 11527 Athens, Greece; 2Department of Pediatrics, University General Hospital of Alexandroupolis, Democritus University of Thrace, 68100 Alexandroupolis, Greece; 3Center for Health Services Management and Evaluation, Department of Nursing, School of Health Sciences, National and Kapodistrian University of Athens, 11527 Athens, Greece; 4School of Medicine, National and Kapodistrian University of Athens, 11527 Athens, Greece; 5Department of Nursing, University of West Attica, 12243 Athens, Greece; 6Department of Nursing, International Hellenic University, 68300 Didimoteixo, Greece; 7Department of Nursing, General Hospital of Kavala, 65500 Kavala, Greece; 8Department of Nursing, School of Health Rehabilitation Sciences, University of Patras, 26334 Patras, Greece

**Keywords:** charter on the rights of children, children rights, human rights, pediatric hospital

## Abstract

*Background and Objectives*: In 1989, the United Nations (UN) General Assembly adopted the United Nations Convention on the Rights of the Child (UNCRC), with a considerable number of the Articles of the Convention being related to the health status of children. Therefore, adhering to and assessing the implementation of the rights of children during hospitalization is a very important step towards child protection. Herein, we attempt to highlight the depth of knowledge of employees working in children’s hospitals with regard to children’s rights as well as the degree of adherence to the UNCRC with respect to hospitalized children. *Material and Methods*: The target group included all healthcare professionals working in the various general pediatric clinics of the three Children’s Hospitals of the Athens metropolitan area in Greece. We conducted a cross-sectional study, with data collection carried out in February and March 2020, using a structured questionnaire consisting of 46 questions which was handed out to all personnel. For the analysis, we used the IBM SPSS 21.0. *Results*: A total of 251 individuals participated in the study (physicians 20%, nurses 72%, and other employees 8%). A total of 54.5% of health professionals did not know what the UNCRC is, and 59.6% of them were not even aware that their hospital had rules and a bioethical committee related to clinical research involving children. Lack of awareness or trust of health professionals is also observed for other procedures or supervisory measures such as abuse protocols, complaint control, admission control, etc. With regard to the health system, there are shortcomings or weaknesses in (a) procedures followed with regard to respect for gender and privacy, (b) information on basic services provided by pediatric hospitals (such as recreation, education and free meals during hospitalization), (c) the logistical infrastructure (such as recreational facilities and facilities for the disabled), (d) the possibility of recording complaints, and (e) hospitalizations that were not necessary. A difference emerged concerning the nurses’ responses between the three hospitals, with nurses participating in relevant seminars held in one of the hospitals being significantly more informed. *Conclusions*: The majority of healthcare personnel seem unaware of basic principles with respect to children’s rights during hospitalization as well as relevant procedures and supervisory measures. Moreover, obvious weaknesses of the health system exist with respect to procedures, services, infrastructure, and complaint recording. There is a need for improved education of health professionals with respect to the implementation of children’s rights in pediatric hospitals.

## 1. Introduction

Determining the optimal living conditions and optimal protection of the child constitutes important cornerstones for their happiness, growth, development, and learning achievements in the family, at school, in the community, and, in general, in life. Historically speaking, for many centuries, childhood was not considered as the unique phase of life we now know it to be [1]. Τhe recognition of the special needs of children and their rights is a fairly recent development. The Geneva Declaration of the Rights of the Child was formulated; this constituted the first international recognition of children’s rights. World War II aroused similar deep sensitivities and concerns; 1947 was the year that saw the creation of United Nations International Children’s Emergency (UNICEF). In 1948, the United Nations (UN) Organization recognized, in Article 25 of the Declaration of Human Rights, that motherhood and childhood are entitled to special care. Finally, in 1989, the United Nations Convention on the Rights of the Child (UNCRC) was unanimously approved by the UN General Assembly and has now been signed by 193 nations [2,3]. Yet another step forward was to ensure optimum conditions of nurturing during the child’s early development (ECD) so that the child may achieve his/her full potential [4]. A further development in these efforts to enhance the welfare of children in hospitals and other healthcare services and, thus, for the overall improvement of their quality of life was the establishment of a charter of rights, setting out the principles of the European Association for Children in Hospitals (EACH-1988) [5].

In certain recent publications, questions have been raised as to whether or not the COVID-19 regulations that prohibit parental visits to their hospitalized children are invalid and whether the ban on visits is a violation of children’s rights. The authors consider that hospitals would be justified in allowing parental visits, provided proper precautions are taken [6,7]. In an analysis by Vance [8], it was found that among 239 children’s hospital guidelines, posted to hospital websites in the USA during one week (June 2020), only 28 did not post a guidelines review. Parental visiting was not prohibited. However, certain restrictions were applied.

In Greece, the UNCRC was embodied in the national legislation in 1992 and is included in a special Article in the Constitution of the Hellenic Republic. A large number of the UNCRC’s Articles on children’s rights (18 out of 54) are, directly or indirectly, related to children’s health, as well as to the manner in which and the degree to which the Articles are complied with in pediatric hospitals, specifically regarding the following: access to health services without discrimination, information offered to families and the opportunity for them and the child to participate in decision making, opportunities for hospitalized children to play and to benefit from recreation and education, and due respect for gender and privacy [9,10]. It is thus of considerable importance to assess the level of knowledge of people working in a children’s hospital with regard to children’s rights and the degree to which these rights are observed. Pertinent data have been published from various countries (Italy, Romania, Iran, and others) in which inadequate possession of relevant knowledge and a lack of observance of children’s rights in hospitals have been reported [5,11,12,13,14,15,16,17]. Analogous data have not, to date, been published from Greece. In the present study, pertinent data provided by people working in the three Athens children’s hospitals, namely, Aghia Sofia Children’s Hospital, Panagiotis & Aglaia Kyriakou Children’s Hospital, and Penteli Children’s Hospital, were collected.

The purpose of this study was to highlight the depth of knowledge of employees working in children’s hospitals with regard to children’s rights as well as the degree of adherence to the UNCRC when children are hospitalized.

## 2. Subjects and Methods

We conducted a cross-sectional study during a period of 2 months (February and March, 2020) in the three pediatric hospitals located in the Athens metropolitan area (i.e., Aghia Sofia Children’s Hospital with 693 beds, Panagiotis & Aglaia Kyriakou Children’s Hospital with 424 beds, and the smaller one, Penteli Children’s Hospital with 160 beds). These hospitals are completely devoted to the healthcare of children, especially the Aghia Sophia Children’s Hospital and the Panagiotis & Aglaia Kyriakou Children’s Hospital, which are located next to each other, forming a large campus, and are the reference centers for complicated and rare disorders for the entire country since, in Greece, no other Children’s Hospitals exist outside of the Athens metropolitan area. The survey had, however, to be discontinued due to the SARS-CoV-2 epidemic. In order to conduct the study, permission was obtained from the scientific and ethical committees of the three hospitals. The participants were informed about the nature and the aim of the study, that their participation was voluntary and anonymous, and that they would need to sign the relevant consent form.

A structured questionnaire consisting of 46 close-ended questions was used in an effort to obtain information related to the participants’ knowledge of the concept of children’s rights and of respecting children’s rights during their hospitalization. Specifically, the questionnaire consisted of three major sections (number of relevant questions provided in parentheses): (a) demographics of participants (6), (b) overall knowledge about UNCRC (3), (c) questions concerning the respect of specific children’s rights during hospitalization, such as the consideration of all dimensions of health (3), unimpeded access to health services (3), necessity of hospitalization (3), opportunity for play and education (5), right to be appropriately informed (3), right to express their opinion (2), right to be protected (2), parental presence (3), privacy (6), dignified death (1), avoidance of unnecessary pain (1), agreed participation in clinical research (2), and expression of complaints (3). For the formulation of the questionnaire, two related questionnaires in English were used, namely: (1) the WHO “Self-evaluation Model and Tool on the Respect of Children’s Rights in Hospitals”, drawn up in cooperation with UNICEF and the UK Healthcare Commission [18], and (2) the questionnaire of the “European Association for Children in Hospital Charter”, as this was presented by Migone et al. [5]. These questionnaires were translated into Greek and were combined into one. Minor modifications were made in terms of wording to better correspond to the Greek reality.

According to the study design, the target group included all healthcare personnel working in the various general pediatrics and surgical clinics of the three above-mentioned pediatric hospitals located in the Athens metropolitan area. The questionnaire was not given to healthcare professionals working in departments where special circumstances and conditions exist, such as the neonatal units, the intensive care units, the psychiatric clinic, the emergency department, day clinics, etc. Furthermore, we conducted a pilot study with 30 participants prior to the final study in order to assess the reliability and the validity of the questionnaire. First, we performed a test–retest study, and we calculated Cohen’s kappa for the items questionnaire since they are categorical variables. Cohen’s kappa for the items ranged from 0.745 to 0.869 (*p* < 0.05 in all cases), indicating a high level of reliability. Additionally, we performed cognitive interviews with the 30 participants in order to assess the face validity of the questionnaire. Only minor changes were applied in the final questionnaire. Nevertheless, the survey abruptly ended due to the lockdown imposed during the SARS-CoV-2 epidemic prior to reaching out to the entire potential study population. Head nurses helped to distribute and collect the completed questionnaires from each pediatric clinic.

## 3. Statistical Analysis

We present categorical variables as numbers and percentages. Additionally, we use the mean and standard deviation to present continuous variables. We used the chi-square test to make comparisons among categorical variables. *p*-values less than 0.05 were considered statistically significant. We used the IBM SPSS 21.0 (IBM Corp. Released 2012. IBM SPSS Statistics for Windows, Version 21.0. Armonk, NY, USA: IBM Corp.) for the analysis.

## 4. Results

Overall, the questionnaire was handed to 353 healthcare professionals, of which 251 responded (response rate 71%). Therefore, the studied population included 251 healthcare professionals working in the three aforementioned children’s hospitals in Athens (physicians 20%, nurses 72%, and others 8%). The demographics of the participants are shown in Table 1.

The responses to questions directly pointing to specific Children Rights and UNCRC Articles, out of a total of 46 questions included in the questionnaire, are displayed in Figure 1 and Figure 2 and in Table 2. Please note that we present categorical variables as numbers and percentages. Additionally, we use the mean and standard deviation to present continuous variables. We used the chi-square test to make comparisons among categorical variables. *p*-values less than 0.05 were considered statistically significant.

Only 45.5% of the participants reported that they were aware of the UNCRC and, specifically, of the Charter on the Rights of Children and Young People applied in hospitals, while very few (6.9%) had observed any pertinent and easily accessible information in the clinics or hospital offices (Figure 1 and Figure 2).

Only 29% of the participants considered that the hospital conforms to the Charter on the Rights of Children and Young People. A total of 38.2% replied that gender and privacy were respected in rooms with more than one bed, while 71.4% stated that curtains were used during the clinical examination and 46.3% stated that there was the potential for a child to be examined by a doctor of the same gender.

To the question of whether there is a separate area in clinics in which privacy during a clinical examination is ensured, 73.7% gave a positive response.

Regarding children’s rights to play, relaxation, recreation, and education in specially designed environments, 44.1% stated that such an area (or areas) for activities exists in the hospital. In total, 79.6% replied that various recreation activities are organized in the hospital, while 25.9% considered that the available opportunities were insufficient. Only 54.2% knew that the hospital provided a specially trained teacher.

Regarding respect for the parents’ and children’s right to have access to information and to the possibility to take part in decision making, 88.3% gave a positive response, while 30.9% considered that they themselves were adequately trained for effective communication with the children and the family.

Concerning the availability and free access to medical services for foreigners, including even those without appropriate certification, 95% replied that this is indeed provided.

In total, 48.8% considered that the hospital required the presence of personnel and volunteers with ethno-cultural skills, although only 3.1% replied positively to the question of whether the teaching of multiculturalism and diversity takes place for the hospital personnel and the volunteers.

Regarding the availability of information related to health in other languages, 37.8% replied positively, 44.4% replied negatively, and 17.8% did not know. Meanwhile, 36.3% stated that bereavement support that is appropriate for individual patients’ cultures is available.

As to the right of the child to be with his/her parents (unless medical reasons prevent it), this was responded to positively by 88.1% of the participants.

Regarding restrictions on the presence of parents in areas where certain medical procedures are carried out, 88.1% consider that adequate explanations are given.

Regarding the question of whether hospitalizations were always necessary, 53.9% replied that a significant number of hospitalizations could have been avoided. The responses did not differ between physicians and nurses.

Only 29.7% considered that appropriate protocols for minimizing substandard care were implemented, partially or entirely, while 22.4% considered that children in the hospital, or the community in general, have easy access to healthcare provisions and/or related information.

Concerning the possibility of food provided, free of charge, to parents staying in the hospital overnight, 73.4% were aware of such a possibility. Regarding the existence of structural obstacles preventing accessibility to certain places for children with mobility problems, 29.8% were aware of such a deficiency.

A total of 59.6% did not know that there were specific regulations and special bioethical committees for clinical research, whereas 12.2% stated that they did not exist. As to whether there have been problems as a result of children’s participation in clinical research, 73.2% replied that they did not know.

Regarding the right of children to pain prevention or management, 63.3% declared that such protocols are used partially or entirely, while 24.3% did not know, and 12.4% stated that they are not followed.

Concerning the use of protocols in the hospital for the protection of children from any form of physical or mental abuse or neglect, 36.7% replied that they are applied entirely, 42.1% replied that they are applied partially, and 21.2% replied that they are applied very little or not at all, while 21.2% did not know. Meanwhile, 41.8% considered that adherence to such protocols is supervised by the hospital, and 43.4% did not know.

Regarding the reporting of complaints, only 14.6% responded that an easily accessible mechanism was provided by the hospital.

To the question of whether written complaints reporting the non-respect of children’s rights could bring about changes, 38.4% replied that they would indeed lead to improvements.

Comparing the responses provided by the three hospitals, it was found that in one hospital (Aglaia Kyriakou), the nurses were aware at a higher percentage (57.7%) of their hospital’s Charter on the Rights of Children and Young People, as compared to the other two hospitals (Aghia Sofia 48.2% and Penteli 34.9%), although the difference was not statistically significant (*p* = 0.08) (Figure 3). In addition, according to the nurses’ responses, the Aglaia Kyriakou hospital follows the Charter on the Rights of Children and Young People at a higher percentage (44.2%) than Aghia Sofia Children’s Hospital (29.4%) and Penteli Children’s Hospital (16.7%) do (*p* = 0.04). The percentage of nurses in Aglaia Kyriakou reporting that they strictly comply with the protocols and procedures related to the protection of children from any form of physical or mental abuse or neglect was 44%, which is significantly higher than the corresponding percentages in the other two hospitals (35% and 33%, respectively) (*p* < 0.001) (Figure 4 and Figure 5).

## 5. Discussion

A large number of Articles of the UNCRC (18 out of 54) relate, directly or indirectly, to children’s health, the primary goal being the provision of the highest possible quality of care so that the child may enjoy the highest attainable standard of health. In summary, these Articles emphasize that it is important for every child to have: (1) Access to health services without any discrimination, (2) Hospitalization when it is really necessary, (3) Opportunity for play and recreation, (4) Education, (5) Avoidance of separation from parents without their agreement, (6) Respect of their privacy, (7) Non-participation in clinical research trials without their permission, (8) Participation in decision making related to their health and, (9) Effort for pain management or prevention.

It must be stressed, however, that providing health in accordance with UNCRC principles requires time and a high level of competence [19]. Informing children and allowing participation in decisions related to their health, taking into consideration their age, understanding, and maturity, will optimize our care. It should be underlined that participation in the decision-making process is placed high in the hierarchy of children’s rights by the children. Specifically, in a study of school children in Geneva [20], participation in decisions was considered by the children as being very important. This decision is shared with children or adolescents with chronic illness [21]. It should be added that a test (the Maturtest) was recently proposed by Miquel E et al. [22] for assessing the minor’s degree of maturity, making them capable of participation in decisions related to their health. Furthermore, the “three good questions” program’s implementation in pediatric medicine to increase shared decision making seems feasible [23].

In the present study, an attempt was made to identify the extent of awareness of information related to children’s rights by the hospital personnel. The results showed that the hospitals’ Charter on Children’s Rights in Hospital was not displayed in areas of the hospital where it could be easily read and that only 6.9% of the responders were aware of its existence. In addition, 29.1% of the participants considered that the hospital conforms to the UNCRC. It was also evident that respect for gender and privacy requires considerable improvement, since only 38.2% replied that gender is taken into consideration in bed distribution in hospital rooms, 71.4% reported that curtains are used to separate beds, and 73.7% replied that there are separate places for physical examinations. It is of note that regarding parents’ and children’s right to be informed, 83.1% of the respondents considered that they offer adequate information to the families, while 81.1% declared that they have appropriate training for communicating with parents and children.

With regard to children’s right to recreation and education in an appropriate area, only 25.9% considered that the existing facilities are adequate. Furthermore, only 54.2% knew that a specially trained teacher is provided by the hospital. It was revealed that arguments frequently arise as to the limited possibility of parents being present in special areas, such as blood collection rooms and neonatal units, although 88.1% of health professionals considered that adequate explanations were given for the procedure followed. Regarding the need for hospitalization, 53.9% replied that a significant number of admissions could have been avoided, there being no difference in the responses between nurses and physicians. This is a phenomenon that particularly warrants evaluation for confirmation and the causes involved. Of special significance is the fact that 29.8% of the participants considered that there are design and construction deficiencies preventing entrance to various hospital areas, requiring appropriate interventions to facilitate the access of children, especially those with a handicap, to areas of recreation and for other activities. Only 14.6% of the participants considered that any easy mechanism exists for the submission of complaints. As for the question regarding clinical research protocols and their supervision by special bioethical committees, an impressive percentage, namely, 59.6%, stated that they were not aware of their existence, while 12.2% reported that they were certain that they do not exist in their hospital. Concerning the protocols aiming to protect children from physical or mental abuse or neglect, 21.2% replied that they did not know about it or that the protocols were not followed, whereas 41.8% considered that this procedure is supervised by the hospital. With regard to the right of the child to have access to pain management or prevention, 63.3% reported that the relevant protocols were applied entirely or partially.

It is of special note that the nurses’ responses differ significantly in two parameters, those of the Aglaia Kyriakou Hospital being more positive. Specifically, 44.2% of nurses of Aglaia Kyriakou replied that the Charter on the Rights of Children and Young People is respected (Figure 4); the corresponding numbers in the Aghia Sofia and Penteli Children’s Hospitals are 29.4% and 16.7%, respectively (*p* = 0.04). An analogous difference between the three hospitals was shown when we compared the nurses’ responses regarding the protection of the child from any form of physical or mental abuse and neglect (Figure 5), the percentages being 44.4%, 35%, and 33.3% in Aglaia Kyriakou, Aghia Sofia, and Penteli Hospitals, respectively (*p* < 0.001). The same trend was observed when responses were given related to awareness among nurses of children’s rights, but the difference was not significant (Aglaia Kyriakou 57.7%, Agia Sofia 48.2%, and Penteli Hospital 34.9%) (*p* = 0.08). Although the reason for these differences is not entirely clear, we attributed the relatively greater awareness among the staff of Aglaia Kyriakou to the fact that seminars on these and other issues regularly take place in the Aglaia Kyriakou Hospital.

The message here, which is a significant one, is that, as in other areas of behavioral instruction, informing and educating all those involved, including the general population, will significantly improve the health and wellbeing and, thus, future prospects of children everywhere. The aforementioned message has also been noted in other studies [24,25]. Literature data from other countries also showed inadequacies in following children’s rights in hospitals, of variable type and degree, the latter most likely related to variability in the study design or the population involved [5,11,12,13,14,15,16,17]. Specifically, in the study of Migone et al. concerning a tertiary referral hospital in Dublin, the findings suggest that healthcare professionals shared similar concerns with the participants in our study, including the lack of appropriate facilities in the hospital for play, education, age-appropriate wards, and the lack of privacy. Interestingly, staff felt that many children undergo unnecessary admission and treatment, as in our study. Moreover, many staff members were reluctant to discuss certain issues such as side effects of medications with patients, do not encourage children to ask questions, and were reluctant to consider children under 16 as capable of giving consent (5). With respect to the issue of parental presence during medical procedures, there are conflicting views and practices. According to the systematic review by Piira et al., although parental presence may not have a clear, direct influence on child distress and behavioral outcomes, there are potential advantages for parents, and clinicians should provide parents with the opportunity to be present during their child’s painful procedure (11). It is interesting that Albert-Lorincz, while investigating pediatric patients’ rights in the Transylvanian healthcare system, found that children are treated by obsolete principles despite the fact that healthcare professionals are knowledgeable with respect to children’s rights. It seems that applying such knowledge, especially changes in practice, in everyday care is a complicated task. Moreover, it seems that difficulties also exist with respect to age-appropriate communication and participation in decision making such that fundamental rights of children are just partially enforced. The author also raises the significant issue of adequate health services being lower for vulnerable children living in poverty (12). In the study of Bisogni and colleagues concerning Italian pediatric units, the most implemented right was the right of children to have their mothers with them, and the least implemented was again the right to express their opinion about care. According to the majority of Italian pediatric nurses, the most important right is the right to pain relief. Significant differences in the implementation of rights were found between areas of Italy and between pediatric hospitals and pediatric units of general hospitals. Overall, according to the perception of pediatric nurses, the implementation of the rights of hospitalized children in Italian pediatrics units is still limited.

It seems, however, that by including children and parents in the study population, a more representative picture will be obtained. It is imperative to create relevant prototypes of care in order to address a large number of problems that are extremely complex and have multiple causes. The following obligations seem to be absolutely necessary.
(1)Respect for gender and privacy(2)Adequate provision of recreation and education facilities; possibility to report complaints(3)Avoidance of unnecessary admissions

Most importantly, parents and children must be adequately informed in accordance with their age, maturity, and mentality and background and, if possible, in their own language.

Achieving these goals requires not only governmental support and laws or conventions but, importantly, the promotion of motivation and education among all those involved in the care of children. A major role in these activities should be played by pediatricians, who need to be trained not only during their residency but also throughout their career by attending relevant seminars. Specifically, they should be trained on how to recognize and report child maltreatment and engage with community systems dedicated to the enhancement of child welfare. Constant efforts to comply with respect for children’s rights in hospitals through the education of all those involved will optimize their care and their quality of life in general [24,25].

## 6. Limitations of Study

The limitations of our study include the fact that we are collecting beliefs and perceptions of healthcare professionals rather than facts and actual actions, that we are not taking into account the time that the healthcare worker has been working in a pediatric environment, and that our study group only includes healthcare professional working in Pediatric Hospitals rather that Pediatric Departments within General Hospitals

## 7. Conclusions

We conclude that significant inadequacies exist in Greek pediatric hospitals with respect to the observance of children’s and young people’s rights during hospitalization, in accordance with analogous findings in similar studies worldwide. The majority of healthcare personnel seem unaware of basic principles with respect to children’s rights during hospitalization as well as relevant procedures and supervisory measures. Moreover, obvious weaknesses of the health system exist with respect to procedures, services, infrastructure, and complaint recording. Moreover, it appears obvious that, as in other areas of behavior modification, the provision of information and education to all those involved, and especially to pediatric nurses as well as to the general population, will ensure that significant progress can be made towards rectifying inadequacies and thereby greatly improving children’s health in their life, which are prerequisites for them reaching their full potential.

## Figures and Tables

**Figure 1 medicina-59-00955-f001:**
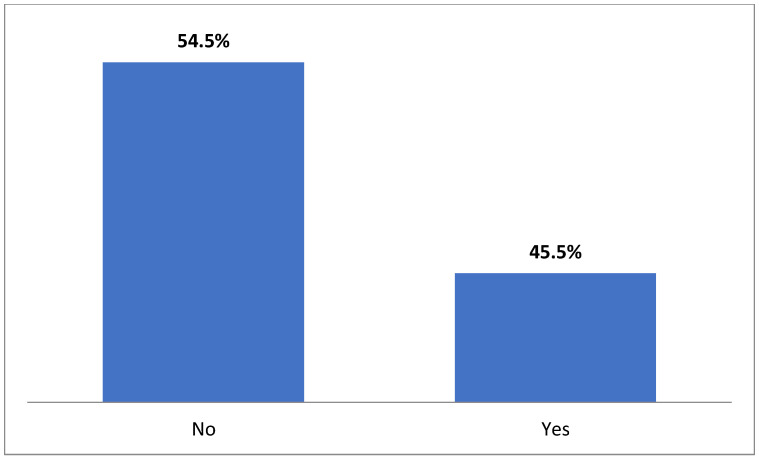
Do you know what the Convention on the Rights of the children is?

**Figure 2 medicina-59-00955-f002:**
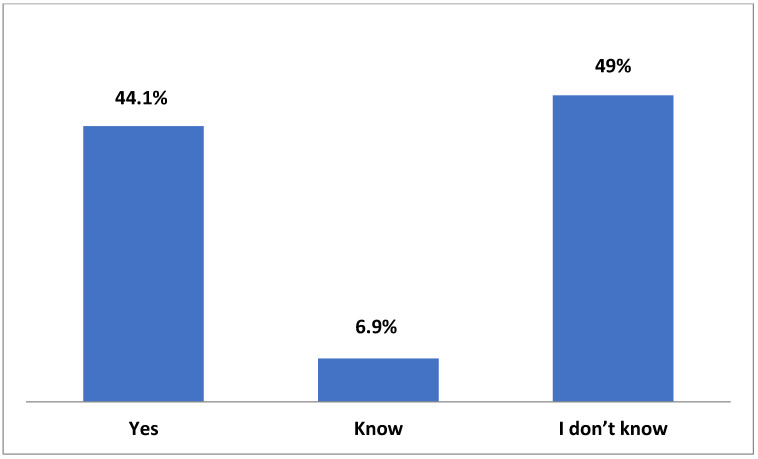
Is the Convention on the Rights of children printed form and visible in the clinics and office of the hospital?

**Figure 3 medicina-59-00955-f003:**
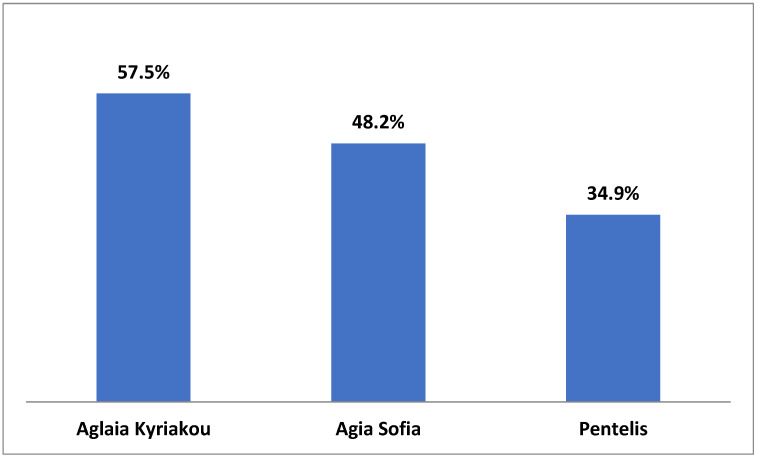
Nurses’ responses to the question: Do you know what the Charter on the Rights of Children and Young People is? Positive response (%).

**Figure 4 medicina-59-00955-f004:**
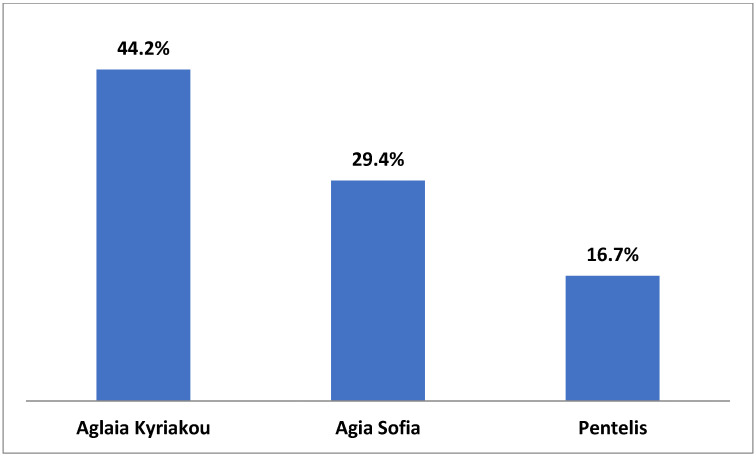
Nurses’ responses to the question: Does your hospital follow the convention of children’s rights?

**Figure 5 medicina-59-00955-f005:**
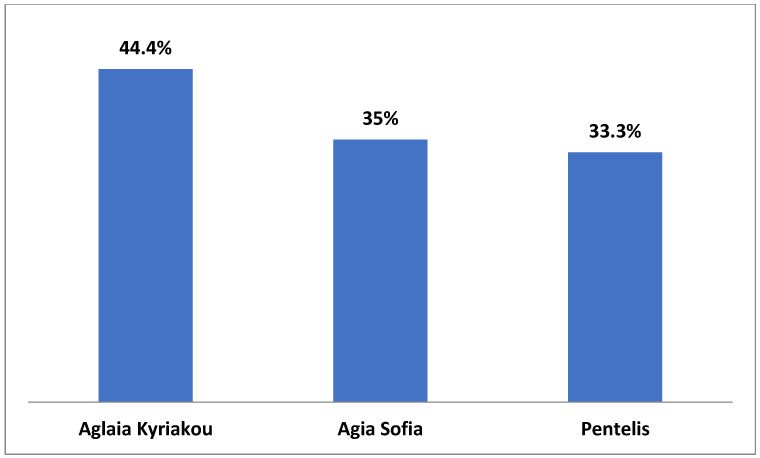
Nurses’ responses to the question: Are the protocols for the protection of children from violence or neglect applied in your hospital?

**Table 1 medicina-59-00955-t001:** Demographics of the healthcare workers included in the study.

**Gender (%)**	
Males	11.1
Females	88.9
Age (years)	41.1 ± 9.1
Prior nursing experience (years)	10.7 ± 13
**Marital status (%)**	
Married	56.5
Single	37.3
Divorced	5.8
**Educational level (%)**	
2 years nursing school	28.4
College	53.2
Master’s/PhD	18.4
**Hospital distribution of participants (%)**	
Aghia Sofia	55.7
Panagiotis & Aglaia Kyriakou	24.1
Penteli	20.2
**Occupation (%)**	
Physicians	20
Nurses	72
Others	8

**Table 2 medicina-59-00955-t002:** Responses given by the participants to questions included in the questionnaire that directly point to specific Children Rights and UNCRC Articles (out of a total number of 46 questions).

Questions		Yes (%)	No (%)	I Don’t Know (%)
1	Is gender respected in rooms with more than one bed?	38.2	45.9	13.4
2	During clinical examinations, were curtains used in rooms with more than one bed?	71.4	23.6	5
3	Is there the possibility for examination by a doctor of the same sex?	46.3	41.4	12.3
4	Is there a separate place for clinical examination and private communication with parents and children?	73.7	20.5	5.8
5	Is there a special place in the hospital for playing, relaxation, recreation, and education?	44.1	54.4	1.5
6	Are various play, recreational, and educational activities organized in the hospital?	79.6	9.6	10.8
7	Are the activities adequate?	25.9	53.9	20.2
8	Does the hospital provide a specially trained teacher?	54.2	29.4	16.4
9	Is pertinent information given to parents and children adequate?	88.3	5.5	6.2
10	Are the health providers adequately trained to communicate with children and family?	30.9	62.2	6.9
11	Is free medical care available to foreigners, including those without appropriate certification?	95	3.1	1.9
12	Does the hospital require the presence of personnel and volunteers with ethno-cultural skills?	21.3	48.8	29.8
13	Is there information related to health in other languages?	37.8	44.4	17.8
14	Is there support available that is appropriate to individuals’ culture?	36.3	21.6	42.1
15	Are parents allowed to stay with the child, unless medical reasons prevent it?	88.1	5.3	6.6
16	Are there adequate explanations given for restrictions on parents’ presence in certain places?	80.9	6.9	12.2
17	Do you consider that hospitalization could have been avoided?	53.9	15.1	31
18	Did parents know of the possibility of food provided by the hospital, free of charge, when staying overnight?	73.4	9.4	17.2
19	Are there structural obstacles preventing accessibility to places for children with mobility problems?	29.8	58.8	11.5
20	Do you know that there are regulations and bioethical committees for clinical research in children?	28.2	12.2	59.6
21	Are you aware of any problems as a result of children’s participations in clinical research?	2.3	24.5	73.2
22	Are there protocols for pain management and are they followed?	63.3	12.4	24.3
23	Are there protocols in the hospital for the protection of children from abuse or neglect?	36.7	42.1	21.2
24	Is there any mechanism for reporting complaints by parents or children?	14.6	54.2	31.2
25	Do you consider that written complaints reporting non-respect of children’s rights would bring about changes?	38.4	16.3	45.3

## Data Availability

All the data generated during this study are included in this published article.

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
