# Peer review of "Assessing the Respect of Children’s Rights in Pediatric Hospitals"

_medicina, 2023, doi:10.3390/medicina59050955_

Round 1
Reviewer 1 Report (Previous Reviewer 1)
The authors have improved the manuscript's quality substantially and their responses are clear and convincing.
Author Response
We would like to thank the Reviewer for appreciating our efforts.
Reviewer 2 Report (New Reviewer)
Interesting concept and I appreciate the novelty of this studying creating the first data set for Greece of this kind to be published. The cross sectional study design and data analysis methods are appropriate for a research paper of this kind. The conclusions lend themselves to how this data can be utilized to further guide increased education to hospital staff to increase compliance with 'children's rights' as defined by the study. All citations were appropriately referenced throughout the text. Only a few points of clarity are below.
1) "Minor modifications were made to better correspond to the Greek reality.", please expand upon what these minor modifications were i.e. are these changes in wording and does this translation alter the questionnaire in a way that is statistically significant?
2. Figure 1-2 shows the combined answers from 'nurse/physician/personnel' vs figure 3-5 showed 'nurse' responses only. Was there a statistically different value of nurse responders or why were only the responses of nurses shown in figure 3-5 vs physicians and other personnel?
3. "Table 2. Response given by the participants to the most important questions included in the questionnaire (i.e., out of a total number of 46 questions)" What is the inclusion criteria for 'most important' and how was that determined?
"The Geneva Declaration of the Rights of the Child was formulated, this constituting the first international recognition of children’s rights" could be changed to "this constituted". Minor syntax errors found throughout but do not make the paper difficult to understand.
Author Response
We would like to thank the Reviewer for the kind words. Suggested changes are highlighted in the text.
1) "Minor modifications were made to better correspond to the Greek reality.", please expand upon what these minor modifications were i.e. are these changes in wording and does this translation alter the questionnaire in a way that is statistically significant?
Indeed, minor modifications were only made in wording without altering the questionnaire in a way that is statistically significant. This is now acknowledged in the text.
- Figure 1-2 shows the combined answers from 'nurse/physician/personnel' vs figure 3-5 showed 'nurse' responses only. Was there a statistically different value of nurse responders or why were only the responses of nurses shown in figure 3-5 vs physicians and other personnel?
Figures 3-5 focus on the major subgroup of participants (nurses, 80% with respect to the whole group). The reason was (1) for homogeneity in order to compare hospitals and (2) because nurses in one of the hospitals (Aglaia Kyriakou) take part in relevant seminars so a direct comparison would make more sense.
- "Table 2. Response given by the participants to the most important questions included in the questionnaire (i.e., out of a total number of 46 questions)" What is the inclusion criteria for 'most important' and how was that determined?
Questions that were not included either focus on demographics, general knowledge about UNCRC or are not directly pointing to specific Children Rights and UNCRC Articles. Wording was changed accordingly in the text since “important questions” does not accurately describe the selection process.
- Comments on the Quality of English Language: "The Geneva Declaration of the Rights of the Child was formulated, this constituting the first international recognition of children’s rights" could be changed to "this constituted". Minor syntax errors found throughout but do not make the paper difficult to understand.
Suggested change was made
This manuscript is a resubmission of an earlier submission. The following is a list of the peer review reports and author responses from that submission.
Round 1
Reviewer 1 Report
The subject of the study is up-to-date and necessary, But the authors have written the article very incomplete and general. With the current format, the article is not worth publishing in this journal.
Abstract
· The second sentence of the introduction of the abstract needs to be rewritten, it should focus on the importance of assessing children's rights.
· The abstract method sub-section is very incomplete, there should be components such as the year of the study, study design, participants, samples, sampling method, data collection tool, and the statistical test used for analysis in brief in this sub-section.
· The conclusion subsection of the abstract section should be written more fully. It is necessary to make a conclusion based on the findings and then refer to the suggestions.
Introduction:
· I don’t any comments.
Methods:
· Provide more complete information about the investigated hospitals, including the number of beds, etc.
· The method section is written very incompletely and vaguely. More complete information about the study design, the research community, the sample, and the sampling method and how to select the samples, the complete introduction of the data collection tool (different sections and the scales used to measure the opinions of health care professionals), how to evaluate the reliability and validity of the questionnaire, the statistical tests used for data analysis, the type of variables and the value determined for the significance of the relationship between the variables should be provided.
Results
· In the method section, it is mentioned that the questionnaire consists of 46 questions, but in the results section, such a case is not noticeable.
· The statistical test used should also be mentioned in the results section.
Discussion:
· The following explanations in the discussion section are not scientific in the current format and should be removed. In summary, it is important for every child to
have:
1) Health services without any discrimination
2) Hospitalization when it is really necessary
3) Play and recreation
4) Education
5) Avoidance of separation from parents without their agreement
6) Privacy
7) Non-participation in clinical research trials without their permission.
8) Participation in decision-making related to their health
9) Pain management or prevention.
· In the discussion section, after presenting the findings of the current research, it is necessary to present more similar studies to compare and express the alignment or discrepancy and its causes. The limitations of the study should be mentioned.
Conclusions:
The first sentence of the conclusion section has no special meaning and should be deleted. A more complete conclusion based on the research findings s
Author Response
We would like to thank the Reviewers for the careful reading of our manuscript, for their valuable time spent and for their suggestions that have improved the presentation of our work. We truly hope that the Reviewers will find the revised version of the manuscript to be appropriate for acceptance.
Please note that all changes are highlighted within the manuscript.
Please find below our detailed answers for all points raised by the Reviewers.
Reviewer 1
- The second sentence of the introduction of the abstract needs to be rewritten, it should focus on the importance of assessing children's rights.
We have added the following phrase according to the Reviewer’s comment:
“Therefore, adhering to and assessing the implementation of the rights of children during hospitalization is a very import step towards child protection”.
- The abstract method sub-section is very incomplete, there should be components such as the year of the study, study design, participants, samples, sampling method, data collection tool, and the statistical test used for analysis in brief in this sub-section.
The abstract method sub-section was re-written according to the Reviewer’s comments and now reads as following:
“The target group included all health care professionals working in the various general pediatric and surgical clinics of the three Children’s Hospitals of the Athens metropolitan area in Greece. We conducted a cross-sectional study with data collection carried in February and March 2020 using a structured questionnaire consisting of 46 questions which was handed out to all personnel. For the analysis we used the IBM SPSS 21.0”.
- The conclusion subsection of the abstract section should be written more fully. It is necessary to make a conclusion based on the findings and then refer to the suggestions.
The conclusion sub-section was re-written according to the Reviewer’s suggestions in order to be based on the findings:
“The majority of health care personnel seem unaware of basic principles with respect to children’s rights during hospitalization as well as relevant procedures and supervisory measures. Moreover obvious weaknesses of the health system exist with respect to procedures, services, infrastructure and complaint recording. There is a need for improved education of health professionals with respect to the implementation of children’s rights in pediatric hospitals”.
- Provide more complete information about the investigated hospitals, including the number of beds, etc.
Additional relevant information were added, as follows:
“We conducted a cross-sectional study during a period of 2 months (February and March, 2020) in the three pediatric hospitals located in the Athens metropolitan area (i.e., Aghia Sofia Children’s Hospital with 693 beds, Panagiotis & Aglaia Kyriakou Children’s Hospital with 424 beds, and the smaller one Penteli Children’s Hospital with 160 beds). These hospitals are completely devoted to the healthcare of children and especially the Aghia Sophia Children’s Hospital and the Panagiotis & Aglaia Kyriakou Children’s Hospital, which are located one next to the other forming a large campus, are the reference centers for complicated and rare disorders for the entire country since, in Greece, no other Children’s Hospitals exist outside of the Athens metropolitan area”.
- The method section is written very incompletely and vaguely.More complete information about the study design, the research community, the sample, and the sampling method and how to select the samples, the complete introduction of the data collection tool (different sections and the scales used to measure the opinions of health care professionals), how to evaluate the reliability and validity of the questionnaire, the statistical tests used for data analysis, the type of variables and the value determined for the significance of the relationship between the variables should be provided.
In line with the Reviewer’s helpful comments we now include more detailed information with respect to our methodology. Specifically, we have added the following:
“We conducted a cross-sectional study during a period of 2 months (February and March, 2020) in the three pediatric hospitals located in the Athens metropolitan area (i.e., Aghia Sofia Children’s Hospital with 693 beds, Panagiotis & Aglaia Kyriakou Children’s Hospital with 424 beds, and the smaller one Penteli Children’s Hospital with 160 beds). These hospitals are completely devoted to the healthcare of children and especially the Aghia Sophia Children’s Hospital and the Panagiotis & Aglaia Kyriakou Children’s Hospital, which are located one next to the other forming a large campus, are the reference centers for complicated and rare disorders for the entire country since, in Greece, no other Children’s Hospitals exist outside of the Athens metropolitan area.
A structured questionnaire consisting of 46 close ended questions was used in an effort to obtain information related to the participants’ knowledge of the concept of children’s rights and of respecting children’s rights during their hospitalization. Specifically, the questionnaire consisted of three major sections (number of relevant questions provided in parentheses): (a) demographics of participants (6), (b) overall knowledge about UNCRC (3), (c) questions concerning the respect of specific children’s rights during hospitalization such as consideration of all dimensions of health (3), unimpeded access to health services (3), necessity of hospitalization (3), opportunity for play and education (5), right to be appropriately informed (3), right to express their opinion (2), right to be protected (2), parental presence (3), privacy (6), dignified death (1), avoidance of unnecessary pain (1), agreed participation in clinical research (2) and, expression of complaints (3). For the formulation of the questionnaire, two related questionnaires in English were used, namely: (1) the WHO “Self-evaluation Model and Tool on the Respect of Children’s Rights in Hospitals” drawn up in cooperation with UNICEF and the UK Healthcare Commission [18] and (2) the questionnaire of the “European Association for Children in Hospital Charter”, as this was presented by Migone et al. [5]. These questionnaires were translated into Greek and were combined into one. Minor modifications were made to better correspond to the Greek reality.
According to the study design, the target group included all health care personnel working in the various general pediatrics and surgical clinics of the three, above mentioned, pediatric hospitals located in the Athens metropolitan area. The questionnaire was not given to health care professionals working in departments were special circumstances and conditions exist such as the neonatal units, the intensive care units, the psychiatric clinic, the emergency department, day clinics etc. Furthermore, we conducted a pilot study with 30 participants prior to the final study in order to assess the reliability and the validity of the questionnaire. First, we performed a test-retest study and we calculated Cohen’s kappa for the items questionnaire since they are categorical variables. Cohen’s kappa for the items ranged from 0.745 to 0.869 (p<0.05 in all cases) indicating a high level of reliability. Also, we performed cognitive interviews with the 30 participants in order to assess the face validity of the questionnaire. Only minor changes were applied in the final questionnaire. Nevertheless, the survey abruptly ended due to the lockdown imposed during the SARS-CoV-2 epidemic prior to reaching out to all of the potential study population. Head nurses helped to distribute and collect the completed questionnaires from each pediatric clinic.
With respect to our statistical analysis we have changed the paragraph as follows: We present categorical variables as numbers and percentages. Also, we use mean and standard deviation to present continuous variables. We used the chi-square test to make comparisons among categorical variables. P-values less than 0.05 were considered as statistically significant. We used the IBM SPSS 21.0 (IBM Corp. Released 2012. IBM SPSS Statistics for Windows, Version 21.0. Armonk, NY: IBM Corp.) for the analysis”.
- In the method section, it is mentioned that the questionnaire consists of 46 questions, but in the results section, such a case is not noticeable.
We have added more relevant information in order not to confuse the reader. Overall, results are provided for the most important questions of the questionnaire including Figures and especially the Table. Specifically, we have added the following phrase in the beginning of the Results section:
“The responses to the most important questions, out of a total of 46 questions included in the questionnaire, are displayed in Figure 1 and 2 and in Table 2”.
Also, we have changed the heading of Table 2 accordingly which now reads as follows:
“Table 2. Response given by the participants to the most important questions included in the questionnaire (i.e., out of a total number of 46 questions)”
- The statistical test used should also be mentioned in the results section.
We have added the following phrase in the Results section according to the Reviewer’s suggestion:
“Please note that we present categorical variables as numbers and percentages. Also, we use mean and standard deviation to present continuous variables. We used the chi-square test to make comparisons among categorical variables. P-values less than 0.05 were considered as statistically significant”.
- The following explanations in the discussion section are not scientific in the current format and should be removed.In summary, it is important for every child to have: 1) Health services without any discrimination, 2) Hospitalization when it is really necessary, 3) Play and recreation, 4) Education, 5) Avoidance of separation from parents without their agreement, 6) Privacy, 7) Non-participation in clinical research trials without their permission, 8) Participation in decision-making related to their health, 9) Pain management or prevention.
The Reviewer is correct that this part of the discussion seems not scientific and out of place in the current format. The paragraph is supposed to summarize the essence of the Articles of the UNCRC that relate, directly or indirectly, to children’s health, which is the focus of our study. We have now moved the paragraph to the introductory paragraph of the Discussion and rephrased as follows:
In summary, these Articles emphasize that it is important for every child to have: 1) Access to health services without any discrimination, 2) Hospitalization when it is really necessary, 3) Opportunity for play and recreation, 4) Education, 5) Avoidance of separation from parents without their agreement, 6) Respect of their privacy, 7) Non-participation in clinical research trials without their permission, 8) Participation in decision-making related to their healt and, 9) Effort for pain management or prevention.
- In the discussion section, after presenting the findings of the current research, it is necessary to present more similar studies to compare and express the alignment or discrepancy and its causes.The limitations of the study should be mentioned.
We now present more similar studies according to the Reviewer’s suggestion, as follows:
“In the study of Migone et al concerning a tertiary referral hospital in Dublin, findings suggest that health care professionals shared similar concerns with the participants in our study, including the lack of appropriate facilities in the hospital for play, education, age-appropriate wards and lack of privacy. Interestingly, staff felt that many children undergo unnecessary admission and treatment as in our study. Moreover, many staff were reluctant to discuss certain issues such as side effects of medications with patients and do not encourage children to ask questions and were reluctant to consider children under 16 as capable of giving consent (5). With respect to the issue of parental presence during medical procedures, there are conflicting views and practices. According to the systematic review by Piira et al, authors conclude that although parental presence may not have a clear, direct influence on child distress and behavioral outcomes, there are potential advantages for parents and that clinicians should provide parents with the opportunity to be present during their child's painful procedure (11). It is interesting that Albert-Lorincz while investigating pediatric patients’ rights in the Transylvanian healthcare system found that children are treated by obsolete principles despite the fact that health care professionals are knowledgeable with respect to children’s rights. It seems that applying such knowledge and especially changes in practice, in every day care is a complicated task. Moreover, it seems that difficulties also exist with respect to age-appropriate communication and participation in decision-making so that fundamental rights of children are just partially enforced. The author also raises the significant issue of adequate health services being lower for vulnerable children living in poverty (12). In the study of Bisogni and colleagues concerning Italian pediatric units, the most implemented right was the right of children to have their mothers with them and the least implemented was again the right to express their opinion about care. According to the majority of Italian pediatric nurses, the most important right is the right to pain relief. Significant differences in the implementation of rights were found between areas of Italy and between pediatric hospitals and pediatric units of general hospitals. Overall, according to the perception of pediatric nurses, the implementation of the rights of hospitalized children in Italian pediatrics units is still limited (13)”.
With respect to the limitations of our study, the following paragraph was added:
“The limitations of our study include the fact that we are collecting beliefs and perceptions of healthcare professionals rather than facts and actual actions, that we are not taking into account the time that the healthcare worker has been working in a pediatric environment and that our study group only includes health care professional working in Pediatric Hospitals rather that Pediatric Departments within General Hospitals”.
- The first sentence of the conclusion section has no special meaning and should be deleted.A more complete conclusion based on the research findings.
We have changed our conclusion according to the Reviewer’s suggestion:
We conclude that significant inadequacies exist in Greek pediatric hospitals with respect to observance of children’s and young people’s rights during hospitalization in accordance with analogous findings in similar studies worldwide. The majority of health care personnel seem unaware of basic principles with respect to children’s rights during hospitalization as well as relevant procedures and supervisory measures. Moreover obvious weaknesses of the health system exist with respect to procedures, services, infrastructure and complaint recording.

Reviewer 2 Report
The manuscript presents originality and a differentiated way of investigating the referred object. We are for approval after minor adjustments indicated.Author Response
We would like to thank the Reviewer for considering our manuscript and the valuable time spent.